# From Diverse Origins to Specific Targets: Role of Microorganisms in Indirect Pest Biological Control

**DOI:** 10.3390/insects11080533

**Published:** 2020-08-14

**Authors:** Frédéric Francis, Hans Jacquemyn, Frank Delvigne, Bart Lievens

**Affiliations:** 1Functional and Evolutionary Entomology, TERRA, Université de Liège-Gembloux Agro-Bio Tech, 5030 Gembloux, Belgium; frederic.francis@uliege.be; 2Laboratory of Plant Conservation and Population Biology, Biology Department, KU Leuven, B-3001 Leuven, Belgium; hans.jacquemyn@kuleuven.be; 3Microbial Processes and Interactions (MiPI), TERRA, Université de Liège-Gembloux Agro-Bio Tech, 5030 Gembloux, Belgium; f.delvigne@uliege.be; 4Department of Microbial and Molecular Systems, Laboratory for Process Microbial Ecology and Bioinspirational Management, KU Leuven, B-3001 Leuven, Belgium

**Keywords:** endophyte, honeydew, microbiome, nectar, phyllosphere, plant-insect interactions, rhizosphere

## Abstract

Integrated pest management (IPM) is today a widely accepted pest management strategy to select and use the most efficient control tactics and at the same time reduce over-dependence on chemical insecticides and their potentially negative environmental effects. One of the main pillars of IPM is biological control. While biological control programs of pest insects commonly rely on natural enemies such as predatory insects, parasitoids and microbial pathogens, there is increasing evidence that plant, soil and insect microbiomes can also be exploited to enhance plant defense against herbivores. In this mini-review, we illustrate how microorganisms from diverse origins can contribute to plant fitness, functional traits and indirect defense responses against pest insects, and therefore be indirectly used to improve biological pest control practices. Microorganisms in the rhizosphere, phyllosphere and endosphere have not only been shown to enhance plant growth and plant strength, but also promote plant defense against herbivores both above- and belowground by providing feeding deterrence or antibiosis. Also, herbivore associated molecular patterns may be induced by microorganisms that come from oral phytophagous insect secretions and elicit plant-specific responses to herbivore attacks. Furthermore, microorganisms that inhabit floral nectar and insect honeydew produce volatile organic compounds that attract beneficial insects like natural enemies, thereby providing indirect pest control. Given the multiple benefits of microorganisms to plants, we argue that future IPMs should consider and exploit the whole range of possibilities that microorganisms offer to enhance plant defense and increase attraction, fecundity and performance of natural enemies.

## 1. Introduction

With a growing intensification of agricultural practices as a means to feed the mounting human population, pests have become a serious burden for crop production [1,2,3,4,5,6,7]. Traditional pest control relies heavily on chemical pesticides and their use has increased dramatically since the 1960s [8]. Although these pesticides were initially very successful, it has become clear that intensive, traditional pesticide use comes with serious economic, social and ecological costs [9], making it socially and ecologically unsustainable.

To meet the various negative effects of chemical pesticides (resistance cases in pests, toxicity on non-target organisms including humans, pollution in environment, …) more and more biopesticides (i.e., a form of pesticide based on microorganisms or natural products) [10,11] and entomophagous arthropods such as predators and parasitoids (i.e., insects laying their eggs on or in the bodies of other arthropods, eventually causing the death of these hosts) are being used to control pest species as part of an integrated pest management (IPM) approach. In general, IPM is defined as a systematic approach to pest management where all possible tactics, including biological control, are implemented to prevent, monitor and control diseases and pests, ensuring that crop damage remains under defined economic thresholds. Chemical pesticides are only used as a last resort when all other tactics fail or are ineffective with care taken to minimize damage to the ecosystem [12,13]. Since January 2014, all European Union (EU) professional growers are obligated to apply IPM tactics according to EU Directive Sustainable Use (Directive 2009/128/EC), indicating that the use of biological control will only increase. Most of these biological control agents are mass-reared arthropods that are released once (referred to as classical biological control, commonly implemented by releasing an exotic natural enemy to control an invasive, non-native pest that has settled in a new geographical area) or multiple times in greenhouses and other crop growth facilities [14]. The latter is known as augmentative biological control, which includes all activities in which natural enemies are periodically released to control native or non-native pest populations. A list of the most commonly used commercial natural enemies deployed in augmentative biological control has been provided by van Lenteren [12]. Hymenopteran parasitoids have been used most extensively in the past because, in comparison to predators, they are more specific, resulting in a more restricted host range and preventing unwanted non-target effects. However, more recently, generalist biocontrol agents such as predatory mites and bugs are increasingly implemented for effective control measures. Despite the obvious benefits of IPM and the increased number of biological control agents commercially available [14], biological control of insect pests is still applied on a very limited scale. In open fields in particular, adoption of biological control agents is limited, and most control practices relate to the stimulation of naturally occurring pest enemies by improving their habitat quality throughout or along arable fields, e.g., by changing small-scale landscape structures and community composition of non-crop plants (e.g., sowing of flower strips, planting of hedge rows, intercropping, …). However, this so-called conservation biological control has recently been shown to have variable success [15] and it remains unclear which structural measures need to be taken to improve pest management in open landscapes [16]. Furthermore, the use of classical biological control is increasingly considered controversial, due to non-target impacts of introducing non-native natural enemies, hindering further implementation of this type of biological control [17]. Introducing non-native species can cause damage to native plants, vector harmful pathogens, cause biodiversity loss, and even replace or interfere with potential native natural enemies [18]. Therefore, there is an increasing demand for effective biocontrol agents that could, alone or in combination with arthropod natural enemies, be used to combat insect pests.

Microorganisms like entomopathogenic fungi, bacteria, nematodes and viruses have been used for the management of insect pests in diverse ecosystems [19]. Entomopathogenic fungi are typically applied as contact insecticides with the expectation of short-term pest control, but long term persistence is also observed depending on habitat conditions. However, despite successful applications, there are several drawbacks associated with the use of such microbial control agents, e.g., susceptibility to environmental stresses, temperature extremes, desiccation and solar radiation [20,21]. To overcome these problems, a general strategy has been to obtain entomopathogens from different geographic regions, or to obtain microbial variants by genetic engineering [22,23]. Another opportunity involves the use of microorganisms for indirect pest control, i.e., by enhancing plant pest resistance and/or by attracting pest natural enemies. Such approach can lead to long-term biological control, especially when combined with other biocontrol agents such as predators or parasitoids [24].

## 2. Microorganisms: Key Elements in Indirect Biological Pest Control

Microorganisms can contribute to plant fitness and plant functional traits in diverse ways [25]. They do not only have the capacity to promote plant growth and reduce the impact of various biotic and abiotic stresses, but may also induce plant defense, deter pest insects, and/or attract natural enemies of pest species [25,26,27,28,29]. Beneficial plant-associated microorganisms can prime the plant defense process, particularly induced systemic resistance (ISR), through different molecular pathways (jasmonic and salicylic acids and/or ethylene) leading to effects on insect pests [30]. Microbe-mediated ISR includes hypersensitive reactions inducing oxidative stress responses that are essential elements of plant defense against herbivores [31]. However, although jasmonic acid signaling is the main ISR pathway against herbivores triggered by root-associated microorganisms, little is known about the tri-trophic level interaction between plants, insects, and microbes [26]. Additionally, although microbes are known to be connected to plants through diverse molecular pathways, their use for indirect biological pest control is still limited and a better understanding of these microbes, their habitat and how they mediate plant–insect interactions is needed (Figure 1). The outcome of plant–microbe–insect interactions can be dependent on the degree of specialization of the insect. Generalist insect herbivores are generally negatively affected by the toxic and deterrent metabolites of a certain plant species, whereas specialist insects are usually not affected, and can even use such compounds to recognize host plants. Studies on *Arabidopsis thaliana* have shown that the specialist caterpillar *Pieris rapae* was not affected by ISR induced by *Pseudomonas fluorescens*, whereas the generalist *Spodoptera exigua* was negatively affected [32]. Below, we discuss a diversity of microorganisms associated to different sources and illustrate how they can affect insect targets, both pests and beneficials, through indirect effects. For direct effects of microorganisms as insecticides the reader is referred to other review articles [13,33].

### 2.1. Rhizosphere Microorganisms

Soil microbiomes are generally complex and contain thousands of interacting microorganisms. A gram of rhizosphere soil might contain around 10^9^ microbial units and 10^6^ distinct taxa [34,35]. Soil microbial mutualists of plants, including mycorrhizal (such as arbuscular myccorhizal or ectomycorrhizal fungi) and non-mycorrhizal fungi (such as *Trichoderma* species) [36,37], and plant growth promoting rhizobacteria (PGPR) (such as *Pseudomonas fluorescens* [38,39]) have been shown to promote plant defense against herbivores both above and below ground [13,40]. Although several soilborne microorganisms have been identified to enhance plant growth or plant strength, or to protect plants against pests and pathogens, so far only relatively few formulated products are commercially available, most probably due to some difficulties to ensure efficacy in the field. Currently, the Organic Materials Review Institute (OMRI) lists 174 products under the category of ‘microbial inoculants’ either as crop fertilizers or as biological agents in crop management tools [41]. Beneficial soil microbes could act in plant–insect interactions in different ways. Firstly, they contribute to plant growing, increasing plant size and vigor resulting in an enhanced food supply for herbivores. Improved nutrient composition also increases the nutritional value of the plants, supporting insect biological performance [42]. Secondly, from a positive point of view soil microbes may induce the production of secondary metabolites and enzymes which confer direct or indirect pest resistance in plants by production of toxic compounds and inducing systemic defense responses, respectively [13]. The final impact on insect performance will depend on the interplay between the benefits derived from the enhanced plant growth and the negative effects derived from the induced resistance. Thirdly, soil microbes can modify the volatile organic compounds (VOCs) release from plants that interfere with plant signaling [40]. Reduced release of aromatics including methyl salicylate and terpenes were observed in *Arabidopsis* after *Pseudomonas* application. An indirect plant defense against the attacking herbivores was associated to the attraction of higher numbers of parasitoids [43]. Additionally, mycorrhizal fungi have been shown to change plant volatiles to attract spider mite enemies [44].

### 2.2. Phyllosphere Microorganisms

The term phyllosphere refers to the total above-ground portions of plants as habitat for microorganisms, particularly leaves, stems, buds and flowers. These microorganisms live both on the surfaces of plant organs or inside plant tissues (endosphere, see below). The phyllosphere microbial community is mainly composed of bacteria, but other microorganisms like fungi, viruses, archaea and algae are also common [45]. Bacteria surpass by far other groups, both in cell numbers and diversity [46]. Most of the bacterial groups are scarcely known or are undescribed species as revealed by recent metagenomics studies [47,48]. Aerial surfaces of plants are often an inhospitable environment for microorganisms, as they are highly influenced by fluctuating abiotic conditions and poor nutrient availability. Nevertheless, several microorganisms have managed to colonize this environment. Successful interactions play a crucial role in the homeostasis of plants and offer benefits like growth promotion, defense against pathogens and in general driving plant performance to cope with different stresses [49,50]. Early studies of phyllosphere microorganisms primarily focused on plant pathogens [51]. However, since most microorganisms are commensal on their host plants, more widespread and deeper studies have been performed [52]. New technologies based on next-generation sequencing have allowed culture-independent analyses to be performed that raised great opportunities for characterizing the phyllosphere microbial diversity and ecological properties. These findings have opened new areas of study that integrate not only plant–microbe relationships, but also their interaction with insects. In the context of suppressing diseases and pests, microbial communities in the phyllosphere have received less attention than those in the rhizosphere. Nevertheless, there is growing evidence that phyllosphere microbial communities can contribute to disease suppression in plants [53,54,55]. There is also some evidence that microorganisms inhabiting leaf surfaces can affect insects. For example, the European corn borer (*Ostrinia nubilalis*) was deterred from ovipositing on maize leaves when *Sporobolomyces roseus*, a common leaf colonizing yeast, was experimentally sprayed on leaf surfaces [56]. The authors concluded that the change in egg laying behavior was probably the result of removal of nutrients by the yeast that are important for the induction of oviposition, chemical deterrence, or interference with adequate access to the leaf surface. Recent studies have shown that leaf and soil microbiomes are linked [57]. Inoculation of distinct microbiomes collected from soils with different plant species altered the leaf microbiome of *Arabidopsis* and resistance of the plants against the caterpillar *Trichoplusia ni* [58]. Further, entomopathogenic fungi like *Beauveria bassiana* and *Metarhizium anisopliae* are not only common in the soil, but some strains also exhibit an endophytic phase that can promote plant growth and insect resistance (see below). Additionally, fungi like *Trichoderma* that were historically considered to be limited to soils are now known to colonize leaves where they can suppress insect pests [59]. More research is needed to uncover how phyllosphere microbiota mediate plant-insect interactions, and how they can be manipulated to protect plants against herbivores.

### 2.3. Endophytes

Endophytes, i.e., fungi or bacteria that occur inside asymptomatic plant tissues, are ubiquitous and form mutualistic associations with diverse plant species, including wild plant species and crop plants. Several entomopathogenic fungi like *B. bassiana, M. anisopliae* and *Lecanicillium lecanii* have been shown to colonize plant tissues as endophytes and have beneficial effects on plant fitness, including resistance against herbivores [60,61]. Increased herbivore resistance is most likely caused by feeding deterrence or antibiosis resulting from the production of endophyte-induced metabolites [61]. Remarkably, these fungi not only provide a benefit to plants by killing their herbivores, but can also translocate nitrogen from aboveground insect cadavers to the plant through fungal mycelia [62]. Endophytic entomopathogenic fungi also protect plants against pathogens, enhance plant growth, increase distribution of soil nutrients within plants, and improve tolerance against abiotic stress and drought [63,64]. These benefits make them highly attractive to be used in agriculture, especially if they can be combined with other biocontrol agents such as parasitoids or predators [65]. However, there is only very little information on the effects of endophytic entomopathogens on higher trophic levels such as parasitoids and secondary parasitoids (i.e., parasitoids of the primary parasitoids; hereafter “hyperparasitoids”), and the few studies performed so far yielded varying results, ranging from no effect to a negative effect on parasitoid life-history traits [66]. Additionally, endophytes have been shown to induce changes in plant odor that led to increased abundance of natural enemies on endophyte-colonized plants, thereby indirectly affecting plant defense [67]. There is also some evidence that endophytic entomopathogens affect the performance and host-selection behavior of hyperparasitoids. For example, parasitized aphid hosts reared on an endophyte-infected model grass (*Lolium perenne*) reduced the lifespan of hyperparasitoids, and hyperparasitoids were able to perceive the disadvantage for their developing offspring in parasitoids from the endophyte environment and can learn to discriminate against them [68].

### 2.4. Insect Microbes

In plants, successful defense relies on fast and specific response to biotic attack. In plant–microbe and plant–insect interactions several elicitors and effectors are recognized by specific and unspecific receptors that trigger signal cascades eventually leading to gene expression and plant responses. Following a review on effector biology [69], effectors can be defined as ‘all pathogen/pest proteins and small molecules that alter plant host-cell structure and function’. The plant defense responses corresponding to the production of herbivore induced plant volatiles (HIPVs) are induced by diverse elicitors such as microbial-, pathogen- and herbivore-associated molecular patterns (MAMPs, PAMPs and HAMPs). HIPVs are highly specific and useful in providing reliable pest information to associated natural enemies [70]. Damage-associated molecular patterns (DAMPs) are plant-derived substances and breakdown products, which indicate tissue damage, whereas HAMPs are molecules from herbivores that come into contact with the plant during feeding [71]. Oral secretions such as regurgitant and saliva play an important role in eliciting plant-specific responses to herbivore attacks. Specific signatures by pests are considered as a combination of particular feeding style (among a diversity of sucking and grinding alimentary behaviour) of the herbivore and the presence of particular HAMPs and as effectors found in their oral secretions [72].

Several HAMPs and other effectors have been found in Lepidoptera saliva, including glucosidase, glucose oxidase (GOX) and various ATPases [73]. The origin of some of them was determined to be not from the insect host itself but from associated microbial symbionts. For example, the saliva and its main component GOX from *Heliothis zea* caterpillars injected with the gut-associated bacterium *Enterobacter ludwigii* played an important role in elevating tomato anti-herbivore defenses. The caterpillar gut-associated bacteria indirectly mediate plant–insect interactions by triggering salivary elicitors [74]. In addition to HAMPs in Lepidoptera saliva, effectors have also been investigated in detail in aphid saliva. *Sitobion avenae* pectinase was found to induce volatile emissions in wheat and attracted parasitoids [75]. Polyphenol oxidase (PPO) from *S. avenae* saliva led to increased expression of genes related to plant defense signaling [76]. Glucose dehydrogenase and oxidase were found in *Myzus persicae* saliva [77]. Several proteins from Chinese gall aphid saliva were found with binding functions and putative effector activity in gall induction [78]. The role of aphid-associated microbes in the regulation of plant defenses seems to be closely related to insect oral secretions. A range of plant defense effectors was identified in aphids even if functional names and origin, insect host or symbionts were still unknown [79,80,81]. Several salivary proteins between 3 and 10 kDa of the green peach aphid can elicit plant defense responses in *Arabidopsis thaliana* [82] but were not fully identified. A saliva comparative analysis from three aphid species revealed that proteins with predicted oxidoreductase and peptidase activities accounted for 43–50% and up to 40% of the total proteins identified in *M. persicae*, *Acyrthosiphon pisum* and *Megoura viciae* saliva. From 5% to 20% of these identifications were related to aphid endosymbionts such as *Buchnera aphidicola* and *Serratia symbiotica* [83].

### 2.5. Nectar and Honeydew Microbes

Many adult parasitoids fulfil their energy requirements by feeding on a broad range of accessible sugar sources such as floral nectar, extrafloral nectar and honeydew. Although these sugar-rich substances have traditionally been regarded as simple food sources for insects, they are also an ideal habitat for particular microbes, thereby driving important ecological interactions between plants, microbes and insects. Floral nectar is commonly inhabited by yeasts and bacteria, whose growth largely depends on their capacity to use scarce nutrients in nectar, withstand high osmotic pressures, and cope with unbalanced carbon-to-nitrogen ratios [84,85]. While most nectar-inhabiting microorganisms have evolved specific adaptations that allow them to survive and proliferate in this harsh environment [86], they are dependent on insect vectors to disperse from one flower to another [86,87]. Nectar microbes have been shown to alter sugar and amino acid composition and concentration, acidity, ethanol concentration, and the amount of secondary metabolites [67]. Furthermore, the metabolic activity of nectar-inhabiting microorganisms also affects nectar temperature [88], and leads to the production of volatiles or fermentation products that contribute to the flavor and scent of nectar [89,90]. These microbe-induced changes in nectar chemistry may in turn affect plant–insect interactions. In general, nectar yeasts have a neutral or positive effect on pollinator preference [91,92,93], and in at least one example, the effect on pollinator preference appears to increase male plant fitness [92]. In contrast, nectar bacteria seem to have negative effects on pollinator behaviour, although only few bacterial species have been investigated so far [94], and bacterial density may play an important role in determining the strength of this interaction [95]. In addition to affecting plant–pollinator interactions, changes in nectar chemistry also affect other ecological interactions. A recent study has shown that specialist nectar yeasts influence the volatile composition of nectar, which in turn led to enhanced attractiveness to nectar-feeding parasitoids [90]. Moreover, it has been shown that nectar foraging by parasitoids can be enhanced by associative learning of nectar yeast VOCs [96]. Furthermore, nectar bacteria have been shown to enhance survival of *Aphidius* parasitoids [97].

Honeydew, i.e., the sugar-rich sticky liquid secreted by aphids and some scale insects when feeding on plant sap, is an important carbohydrate source for parasitoids, increasing parasitoid longevity and fecundity, especially in agro-ecosystems that often lack alternative sugar sources [98,99]. Honeydew also serves as a host-finding cue for parasitoids and predators of honeydew producers [100,101]. In this regard, honeydew-inhabiting microorganisms have been found to play an important role in determining prey location and ovipositional preferences [102]. Both crude and *Staphylococcus sciuri*-inoculated honeydews strongly stimulated oviposition of hoverflies, important enemies of aphids, and gave an identical number of eggs than obtained with the positive control (plants infested with aphids). By contrast, Goelen and colleagues [103] showed a negative relationship between honeydew bacteria and attraction of the aphid parasitoid *Aphidius colemani*, suggesting that VOC composition and parasitoid response is context-dependent. While aphid honeydew is primarily considered to comprise sugars and amino acids, its protein composition has only recently been investigated and found to be highly diverse. This huge diversity in proteins originates from several organisms, including the aphid host, endosymbiotic bacteria and gut flora, and is composed of several proteins that might act as mediators in the plant–aphid interaction [104].

It is generally believed that insects respond to microbe-mediated VOCs in nectar and honeydew as they serve reliable signals for a suitable habitat or sugar-rich food source for the insects [105]. Additionally, the VOC producing microbes may also provide the insects other benefits such as aid in food digestion or improvement of insect health, e.g., by protection against diseases [106]. Furthermore, consumption of microbes may provide the insects with essential nutrients improving overall insect fitness. Although the precise mechanisms are not fully understood, enhanced attraction of insect vectors may allow the otherwise immotile microorganisms to be dispersed to new habitats [107]. Likewise, some microbial pathogens have been found to alter the host plant’s volatile profile to attract the pathogen’s insect vector [108,109]. Further research is needed to unravel the precise role of these microbes in plant–insect interactions, and how they can be applied efficiently to suppress pest insects.

## 3. Production and Formulation

Whereas many microbes from diverse origins may have potential for indirect biocontrol of pest insects (see Table 1 for some case studies), several aspects should be considered before implementation and broad use. One important challenge is the up-scaling of microbial production based on adapted technologies to improve production efficiency to be easier and less expensive. Efforts are being made to optimize microbial growth on grain residues (such as for *Metarhizium*, *Beauveria*, and *Aspergillus* fungi) associated to innovative biofilm techniques or in bioreactors with best adapted media and conditions (such as for *Bacillus*, *Pseudomonas* and other bacteria) [110]. New strategies for pest biocontrol may come from the microorganisms from the insects themselves (associated to endosymbionts or gut microbes). Therefore, innovative methods for microbial isolation and culturing should be developed to increase the understanding of these microbiota globally, but also individually at the level of microbial species. The dependence on in vivo insect cells due to the closed interactions and the way to provide similar conditions to grow in bioreactors are the main characteristics to elucidate for further efficient and larger production. Microbes in the digestive tract of insects that significantly affect host physiology, ecology and evolution should be considered as a key player in shaping insect-plant interactions [111] and seem a highly promising source to control pests. Further improvements should also target the kinds of formulations to be adapted to microbial insecticides. Depending on the active material, either the microorganism itself (or particular resistance forms such as spores) or molecules/secondary metabolites produced in the growing medium, different formulations should be investigated and adapted: either as liquid or solid after drying leading to emulsion, encapsulation or granule shapes. The broader availability of microbial biopesticides for innovative pest control is directly related to the capacity to adapt and propose optimal formulations. For example, some endophytes have to be applied on roots and some others on leaves to be fully active and will then orient different formulations. Also, plant nectar and insect honeydew microbes have to be dispersed on plant aerial parts, leaves and flowers, and therefore need to be sprayed in liquid applications. Further concerns to include the survival and stability of the microorganisms to ensure long term effects, mainly when VOCs are the active compounds from living microorganisms. Moreover, even if one particular pest is targeted when using a microbial formulation, another consideration will be to consider non-target organisms constituting the entomofauna including beneficials such as entomophagous predators/parasitoids and pollinators. It may be expected that by overcoming these challenges a new range of ecofriendly products will become available to combat insect pests and become integrated in IPM programs.

## Figures and Tables

**Figure 1 insects-11-00533-f001:**
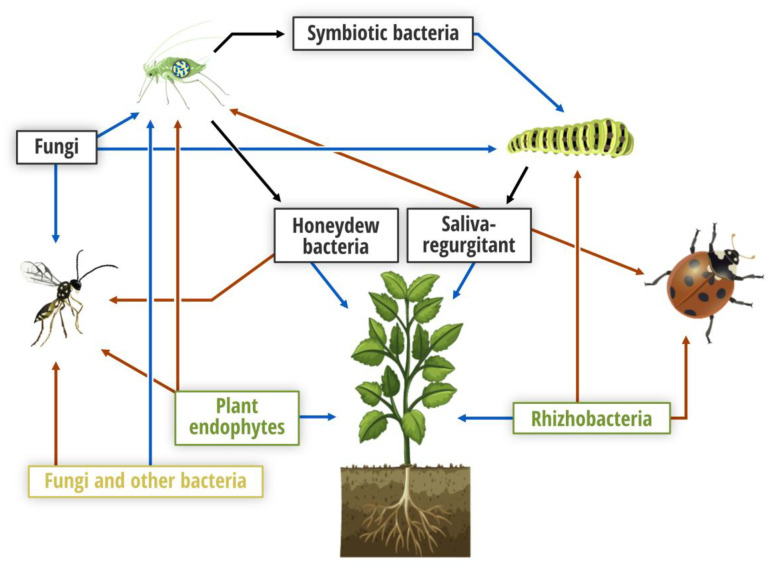
Illustration of multi-trophic plant–pest interactions mediated by microorganisms focusing on plants, herbivores and entomophagous arthropods. Black, green and orange colors refer to insect, plant (rhizo-, phyllo- and endospheres) and herbivore sources, respectively. Blue and red arrows indicate direct and indirect (through plants) effects on target organisms, respectively, while black arrows illustrate insect origin.

**Table 1 insects-11-00533-t001:** Examples of case studies of tri-trophic interactions between microorganisms, herbivores and their natural enemies pointing out an indirect biocontrol effect of microbes.

*Soilborne Microbes*	*Pests*	*Beneficials*	*Effects*	
*Pseudomonas fluorescens WCS417r*	*Myzus persicae* Sultzer	*Diaeretiella rapae* M’Intosh	Increasing parasitoid attraction to plant	[112]
*Pseudomonas fluorescens WCS417r*	*Mamestra brassicae* L.	*Microplitis mediator* Haliday	Plant growth and parasitoid attraction	[38]
*Enterobacter aerogenes*	*Spodoptera littoralis* Boisduval	*Cotesia marginiventris* Cresson	Increasing parasitoid attraction to plant	[113]
*Pseudononas syringae*	*Myzus persicae*	*Coccinella septempunctata* L.	Plant pest resistance and parasitoid attraction	[114]
***Endophytes and plant symbionts***				
*Beauveria bassiana, Metarhizium brunneum*	*Myzus persicae*	*Aphidius colemani* Vierick	Reduction of pest development and parasitoid biology	[24]
*Aspergillus flavus, A. niger*	*S. littoralis*	*Bracon hebetor* Say	Pest resistance and beneficial attraction	[115]
*Neotyphodium lolii*	*Agrotis ipsilon* (Hufnagel)	*Copidosoma bakeri* (Howard)	Negative development of parasitoid	[116]
*Beauveria bassiana*	*Aphis gossypii* Glover	*Chrysoperla carnea, A. colemani*	Reductiion of pest and beneficial performances	[117]
***Nectar and honeydew microbes***				
*Metschnikowia gruessii, M. reukaufii*	Not studied	*Aphidius ervi*	Attraction and dispersal of parasitoids	[90]
*Bacillus* strains	Not studied	*Aphidius colemani*	Attraction of parasitoids	[103]
*Staphylococcus sciuri*	*Acyrthosiphon pisum*	*Episyrphus balteatus* Degeer	Attraction and longevity of parasitoids	[102]
***Insect microbes***			
	Still no evidence on both trophic levels

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
