# Peer review of "From Diverse Origins to Specific Targets: Role of Microorganisms in Indirect Pest Biological Control"

_insects, 2020, doi:10.3390/insects11080533_

Round 1

Reviewer 1 Report

Dear authors,

The manuscript is reviewing a list of subjects related to the role of microorganisms in plant health. The review stress an interesting view of the broader role of microorganisms not only as control agents of pests and diseases but also as modulators of pests and other arthropods behavior and physiology. I find the different perspectives refreshing and interesting.

However, there is a lack of methodological presentation of the subjects and the terms the authors use and I recommend to better define and refer to the terminologies. It will be important first to define the terms of biological control and IPM separately. That way the authors will not confuse the reader and will not misuse each of the terms. Than it is needed to define the broader aspects that the review is presenting. Meaning, defining plant resistance enhancement and separating it from the phenomena of direct enhancement of beneficial organisms such as natural enemies. Only than the authors can suggest new terms such as 'indirect pest biological control'.

Some specific comments:

Abstract:

Line 15: Integrated Pest Management (IPM) is not a decision support system, but an approach  

Line 18: The authors claim that most biological control programs rely on microbial agents. This claim is not correct. I advise the authors to read recent literature on the subject.

Line 20: It seems that the authors claim that plant defense against herbivores is not exploited as a component in IPM. In my opinion, based on several studies in the field of IPM, plant defense is already well establish component in IPM

Introduction:

  • I find some of the citations not completely reflecting the intention of the authors. Some citations are also relatively not updated. For example, Oerke 2006 is an excellent review but it has been 16 years since that data was gathered. I suggest to add newer literature as well. Another example, references 2-5 deal all with the dispersion of fungal pathogens. It is advised to add references regarding pests.
  • The authors refer to natural enemies as the key-players in biocontrol. Add information also regarding biopesticides which are far more exceeding the natural enemies in terms of global trends and commercialization. See recent reviews published in Journal of Invertebrate Pathology Volume 165, 2019 by several authors.
  • Line 79: The authors are presenting use of entomopathogens but the whole paragraph is full with mistakes. It is mainly because of misuse of references. The authors are not describing the whole picture but rather a very narrow one. Typically applied as contact insecticides is correct only for entomopathogenic fungi. There are many examples of successful application of entomopathogens in diverse ecosystems, mutagenesis is not a widely accepted strategy….and again authors use rather old literature to justify these claims.
  • line  88 "Another strategy involves the use of microorganisms for indirect pest control, i.e. by enhancing pest resistance…."  How that improve IPM ?

Reference:

There are duplications in the reference list and some in correct years of publication.

Author Response

Abstract:

Line 15: Integrated Pest Management (IPM) is not a decision support system, but an approach  

Change was brought.

Line 18: The authors claim that most biological control programs rely on microbial agents. This claim is not correct. I advise the authors to read recent literature on the subject.

Some more references were consulted and added.

Line 20: It seems that the authors claim that plant defense against herbivores is not exploited as a component in IPM. In my opinion, based on several studies in the field of IPM, plant defense is already well establish component in IPM

Sentence was adapted.

Introduction:

  • I find some of the citations not completely reflecting the intention of the authors. Some citations are also relatively not updated. For example, Oerke 2006 is an excellent review but it has been 16 years since that data was gathered. I suggest to add newer literature as well. Another example, references 2-5 deal all with the dispersion of fungal pathogens. It is advised to add references regarding pests.

References were added :

Savary, S.; Willocquet, L.; Pethybridge, S.J.; Esker, P.; McRoberts, N.; Nelson, A. The global burden of pathogens and pests on major food crops. Nature Ecology and Evolution 2019, 3, 430-439.

Lehmann, P.; Ammunet, T.; Barton, M.; Battisti, A.; Eigenbrode, S.D.; Jepsen, J.U.; Kalinkat, G.; Neuvonen, S.; Niemela, P.; Terblanche, J.S.; Okland, B.;  Björkman, C. Complex responses of global insect pests to climate warming. Frontiers in Ecology and the Environment 2020, 18, 141-150.

  • The authors refer to natural enemies as the key-players in biocontrol. Add information also regarding biopesticides which are far more exceeding the natural enemies in terms of global trends and commercialization. See recent reviews published in Journal of Invertebrate Pathology Volume 165, 2019 by several authors.

References were added :

Arthurs, S.; Dara, S.K. Microbial biopesticides for invertebrate pests and their markets in the United States. Journal of Invertebrate Pathology 2019, 165, 13-21.

Hatting, J.L.; Moore, S.D.; Malan, A.P. Microbial control of phytophagous invertebrate pests in South Africa: Current status and future prospects. Journal of Invertebrate Pathology 2019, 165, 54-66.

  • Line 79: The authors are presenting use of entomopathogens but the whole paragraph is full with mistakes. It is mainly because of misuse of references. The authors are not describing the whole picture but rather a very narrow one. Typically applied as contact insecticides is correct only for entomopathogenic fungi. There are many examples of successful application of entomopathogens in diverse ecosystems, mutagenesis is not a widely accepted strategy….and again authors use rather old literature to justify these claims.

Misuse of references was checked and corrected.

One reference was added :

Ruiu, L. Microbial biopesticides in agroecosystems. Agronomy 2018, 8, 235.

  • line  88 "Another strategy involves the use of microorganisms for indirect pest control, i.e. by enhancing pest resistance…."  How that improve IPM ?

Sentence was adapted.

Reference:

There are duplications in the reference list and some in correct years of publication.

References were checked and adapted.

Reviewer 2 Report

An excellent review of a topic gaining increasing interest.  Summarizes the field very well with appropriate references.  Nicely written and reading flows from on topic point to the next.

Will be of interest to many in this field of study, and other researchers will use as a foundation to their own manuscripts to follow. Figure was well done, easy to understand, clear, and adds to the topic.

Reference list was one of the best I had reviewed following format requirements, well done.

Only 5 locations in text could delete a word, and one location looked as if the reference number is needed as looks like a partial or typo?

Line 310   has a '1'  ?

Line 153 delete word  'play'

line 193 delete word REF, or add REF  number;

Line 298 could delete 'Only' and Start sentence with 'Few...."

Line 322 delete 'have also'  and change wording 'to be integrated 'to include' survival    so delete word 'on'

see highlighted lines in manuscript, for these areas that need attention. 

Author Response

An excellent review of a topic gaining increasing interest.  Summarizes the field very well with appropriate references.  Nicely written and reading flows from on topic point to the next.

Will be of interest to many in this field of study, and other researchers will use as a foundation to their own manuscripts to follow. Figure was well done, easy to understand, clear, and adds to the topic.

Reference list was one of the best I had reviewed following format requirements, well done.

Only 5 locations in text could delete a word, and one location looked as if the reference number is needed as looks like a partial or typo?

  1. Line 310   has a '1'  ?

It was deleted.

  1. Line 153 delete word  'play'

It was deleted.

  1. line 193 delete word REF, or add REF  number;

It was deleted.

  1. Line 298 could delete 'Only' and Start sentence with 'Few...."

It was adapted.

  1. Line 322 delete 'have also'  and change wording 'to be integrated 'to include' survival    so delete word 'on'

It was changed.

  1. see highlighted lines in manuscript, for these areas that need attention. 

Everything was taken into account for reviewer 2.

Round 2

Reviewer 1 Report

Dear authors,

Some minor corrections are needed:

line 81-82 : "Entomopathogenic fungi are typically applied as contact insecticides with the expectation of short-term pest control". That is not entirely correct, see papers discussion long term persistence and activity of EPF. That depends greatly on habitat conditions. rewrite.

line 85-87: please add to the various strategies also definition of the appropriate application period (seasonal, evening application etc..), and targeting the more susceptible and more exposed life stage of a pest.

Author Response

Please find enclosed the corrected paper according to the reviewer minor revision one (in red in the text).
For second minor revision, I do not understand what and where the reviewer want an adding. I propose to let this part as before because it will not have added value and will make the text less fluent to read.  
